# Factors influencing inappropriate use of antibiotics: Findings from a nationwide survey of the general public in Malaysia

**Li Ping Wong**[1]*, **Haridah Alias**[1], **Suraya Amir Husin**[2], **Zawaniah Brukan Ali**[2], **Benedict Sim**[3], **Sasheela Sri La Sri Ponnampalavanar**[4]*

1 Centre for Epidemiology and Evidence-Based Practice, Department of Social and Preventive Medicine, Faculty of Medicine, University of Malaya, Kuala Lumpur, Malaysia, 2 Infection Control Unit at Medical Development Division, Ministry of Health Malaysia, Putrajaya, Malaysia, 3 Infectious Diseases Unit, Department of Medicine, Sg. Buloh Hospital, Sungai Buloh, Malaysia, 4 Department of Medicine, Faculty of Medicine, University of Malaya, Kuala Lumpur, Malaysia

* wonglp@ummc.edu.my (LPW); sheela@ummc.edu.my (SSLSP)

**Data Availability Statement:** All data files are available in the Kaggle database www.kaggle.com/dataset/

## Abstract

Antibiotic resistance is one of the biggest threats to global public health. Misuse of antibiotics has never been investigated on a nationwide scale among the general public in Malaysia. This study aimed to identify sociodemographic and knowledge factors associated with inappropriate use of antibiotics in the Malaysian context to inform the development of interventions to mitigate inappropriate antibiotic use. We conducted computer-assisted telephone interviews (CATI) between June 2019 and December 2019. The telephone numbers were randomly generated from the electronic residential telephone directory of all 13 states and 3 Federal Territories in Malaysia. The survey consisted of questions on demographics, knowledge about antibiotics and antibiotic resistance (53 items), and practices of antibiotic use (16 questions). A total of 864 complete responses were received. Pronounced erroneous beliefs that antibiotics are effective against infections caused by viruses and that antibiotics can speed up recovery from coughs and colds were evident. The proportions that were aware of the terms 'drug resistance', 'antimicrobial resistance', and 'superbugs' were low. The mean and standard deviation (SD) for the antibiotic knowledge score was 23.7 (SD ±9.25; range 0 to 50) out of a possible score of 52. Regarding antibiotic practices, a considerable proportion reported non-adherence to recommended doses. The mean and SD for the antibiotic practices score was 37.9 (SD ± 6.5; range 17 to 47) out of a possible score of 48. Participants who earned an average monthly household income of MYR1001-3000 (OR 1.61, 95% CI 1.04–2.50) were more likely to report higher antibiotic practice scores than those with <MYR1000. Participants with tertiary education attainment reported higher antibiotic practice scores (OR 1.99; 95%CI 1.02–3.91) than those with primary school and below. High antibiotic knowledge scores (OR 3.94; 95% CI 2.71–5.73) were associated with higher antibiotic practice scores. Inappropriate antibiotic use is influenced by demographics and antibiotic knowledge. This study calls for education interventions focused on the lower socio-economic status population to increase awareness and to promote appropriate antibiotic use.

## Introduction

Antimicrobial resistance is a public health concern around the world that significantly impacts patient health and poses a serious threat to the future health and welfare of both humans and animals [1]. It is well-recognised that inappropriate antibiotic use is an important and modifiable driver of antibiotic resistance [2, 3]. The consequences of antibiotic resistance are numerous, ranging from increased healthcare costs, increased morbidity and mortality, and reduced effectiveness of health delivery services [4]. According to the Centres for Disease Control and Prevention (CDC), an estimated of 2.8million people are infected with antibiotic-resistance bacteria or fungi, and more than 35,000 people die as a result [5].

Toward the aim of promoting appropriate antibiotic use, the World Health Organisation (WHO) has called for action to improve antimicrobial awareness and promote behavioural change through effective communication and education [6]. Hence, public awareness of antibiotic resistance is crucial to mitigating this insidious problem. It is well-established that poor knowledge regarding antibiotic usage gives rise to inappropriate consumption of antibiotics, causing the emergence of resistant microorganisms [7]. Various forms of inappropriate use of antibiotics, such as incomplete of the entire course of antibiotic treatment, skipping doses, using leftover antibiotics from previous treatment, misuse of antibiotics to cure viral infections, and self-medication or purchase of antibiotics without prescription have been commonly reported [8].

The role of the public in tackling antimicrobial resistance is undeniably important. Despite this, there are still few studies addressing the level of knowledge and awareness on antibiotic resistance in the Malaysian context. As knowledge about the risk associated with misuse of antibiotics is the key to dealing with antibiotic resistance, understanding the level of knowledge on antibiotics and the practice of antibiotic use is crucial for providing insight into designing educational interventions. There have been several small-scale studies conducted in Malaysia, all of which suggest that local communities exhibit gaps in antibiotic awareness and appropriate antibiotic use [9–11]. Furthermore, most of these studies have been limited by small sample size. Nonetheless, large-scale population-level data are lacking. Furthermore, comprehensive antibiotic-related knowledge and practices in Malaysia have not been reported. In view of this, this study aimed to investigate a wide range of knowledge on antibiotic usage and antibiotic resistance, as well as inappropriate antibiotic practices at a nationwide level.

## Materials and method

### Participants recruitment

We conducted a cross-sectional nationwide survey of the general population in Malaysia. Telephone interviews were conducted between June 2019 and Dec 2019 using a computer-assisted telephone interview (CATI) system. The telephone numbers were randomly generated by computer from the latest electronic residential telephone directory (2018/2019) of all 13 states and 3 Federal Territories in Malaysia to obtain a representative sample. The sample size was calculated using the equation: $n = Z2\ p(1-p)/d2$. Using a margin of error of 0.05 (5%), with a 95% CI and 50% response distribution, the calculated sample size was 384. Due to the use of convenience sampling, the sample size was multiplied by the predicted design effect of two. Therefore, the minimum survey sample size was set to 768 (384 x 2) participants. Based on an estimate of 10% response rate of telephone survey [12], approximately 11,000 phone numbers were randomly selected from the telephone directory. Inclusion criteria were: Malaysian citizen aged above 18 years old, had heard or known of antibiotics, and residing in the contacted household. In each contacted household, only one person was surveyed. If more than one person in the

contacted household met the inclusion criteria, one person was randomly selected to respond to the survey using. A random number table was used to select the participant to be included in the survey. To avoid over-representation of unemployed participants, all interviews were conducted between 5:30 pm and 10:00 pm on weekdays and from 12:00 p.m. to 7:00 p.m. on weekends or public holidays. Attempts to call at least two times on separate days we made for unanswered calls in the contacted household before being regarded as non-responses.

## Instruments

The survey questionnaire consisted of three sections assessing: 1) socio-demographic characteristics, 2) antibiotic and antibiotic resistance-related knowledge, and 3) practices regarding antibiotic use.

Questions on knowledge about antibiotic usage and antibiotic resistance consisted of six sections (52 items); i. general information about antibiotics (5 items), ii. antibiotic usage (7 items), iii. diseases that can be treated and cannot be treated with antibiotics (11 items), iv. antibiotic side effects (10 items), v. antibiotic resistance-related terms (5 items), and vi. antibiotic resistance problems (14 items). For each statement, the options answer were *yes*, *no* or *don't know*. The correct response was given a score of one, and an incorrect or *don't know* was scored as zero. Eight items where the correct response is false were reverse coded. Antibiotic knowledge scores were calculated by adding up the score on the 52 items. The possible score range between 0 and 52, with a higher score indicates a higher level of antibiotic knowledge. Practices of antibiotic use consisted of 16 questions; i. obtaining antibiotics (3 items) ii. antibiotic consumption and dosing (6 items), and iii. use of leftover antibiotics (7 items). For each question, the response options were *never*, *rarely*, *sometimes*, and *often*, scored as 0, 1, 2, and 3, respectively. Negatively worded practices were reverse scored. The possible antibiotic practice scores ranged from 0 to 48, where higher scores implied a greater level of appropriate antibiotic use. The 52 items for knowledge and 16 items of the practice questions had reliability (Cronbach's α) of 0.931 and 0.880, respectively.

All the questions were developed in reference to previous literature and guideline [13, 14] by the research team and validated by panel of experts that consisted of physicians, academicians, and veterinarians from government agencies. The questionnaire was developed in English. As the largest group of Malaysians consists of three main ethnicities, namely the Malays, Chinese, and Indians, the questionnaire was translated into Bahasa Malaysia (the national language of Malaysia), Chinese (Mandarin), and Tamil by native-speaking translators. The translated questionnaires were reviewed byindependent translators and back translation was conducted on the primary translated version. The questionnaire was pilot tested on random samples of the different ethnic populations from the telephone directory. A team of trained interviewers from three ethnic groups performed the interviews and each interviewer was assigned to interview respondents of a similar ethnic group. The interviewers were trained to reduce interviewer-related errors by ensuring that all respondents were asked identically worded questions without unscripted commentary that could bias response. Participation was voluntary and the participants provided verbal informed consent before the start of the telephone interview. They were also informed that completion of the telephone interviewed implied consent to participate.

## Ethical approval

The study was approved by the University of Malaya Research Ethics Committee (UM.TNC2/ UMREC—531). Informed consent was obtained verbally before the commencement of the telephone interview. This method of consent was approved by the ethics committee.

## Statistical analyses

Descriptive statistics were used to describe the proportion of knowledge and antibiotic practices. The reliability of the knowledge and practices items were evaluated by assessing the internal consistency of the items representing the scores. The normality of total knowledge and practices scores were checked using the Kolmogorov-Smirnov test. Dichotomization of total knowledge and practices scores were done using median split to form high and low score groups [15]. Multivariable logistic regression analysis was used to determine the demographic factors influencing the level of knowledge. Multivariable logistic regression for the outcome variable antibiotic practices included demographic characteristics and level of knowledge. The method allows us to assess the impact of multiple covariates including the demographic variable in the prediction model. The outcome variable level of knowledge and antibiotic practices in the multivariate analyses we divided into two groups for comparison based on the median split. In the multivariable logistic regression analyses, all variables found to have a statistically significant association (two-tailed, $p$-value $< 0.05$) in the univariate analyses were entered into the model via forced-entry method. Odds ratios (OR), 95% confidence intervals (95%CI), and $p$-values were calculated for each independent variable. We conducted causal mediation analysis to investigate the effect of confounder bias on the casual inference of the mediator to the outcome variables using regression analysis. A $p$-value of less than 0.05 was considered statistically significant. All statistical analyses were performed with the Statistical Package for the Social Sciences Version 20.0 (SPSS; Chicago, IL, USA).

## Results

### Participant characteristics

A total of 11,036 randomised numbers were contacted and 1,005 people responded. Of these, only 864 (86.0%) had heard of or were aware of antibiotics and proceeded with the survey. The demographic characteristics of the 864 participants are shown in Table 1. The mean age was 47.3 years (standard deviation [SD] 18.0; range 18–89). The study had a slightly higher representation of female participants (67.8%, n = 586), participants with tertiary education attainment (49.5%, n = 428), and with an average monthly household income of MYR1001 to 3000 (47.8%, n = 413). By ethnicity, the majority were Malay (51.5%, n = 445).

### Antibiotic and antibiotic resistance-related knowledge

S1 Fig shows the correct responses of knowledge items. The majority had good knowledge of antibiotic usage and antibiotic resistance problems, where over 50% of participants provided correct responses on the question items. A small proportion was aware that antibiotics are not effective against infection caused by viruses (23.8%, n = 206) and that antibiotics cannot speed up recovery from coughs and colds (23.0%, n = 199). Additionally, a small proportion was aware that cold and flu cannot be treated by antibiotic (28.7%, n = 248). Awareness of terms such as 'drug resistance' (12.2%, n = 105), 'antimicrobial resistance' (6.0%, n = 52), and 'superbugs' (5.9%, n = 51) was extremely low. Knowledge regarding diseases that can or cannot be treated with antibiotics was also poor.

The mean for the antibiotic knowledge score was 23.7 (SD ±9.25; range 0 to 50) out of a possible score of 52. The median score was 23.0 (interquartile range [IQR], 17.0 to 29.0). On checking the normality distribution of the antibiotic knowledge score using the Kolmogorov-Smirnov test, it was found that the data were not normally distributed ($p$ <0.05). The knowledge scores were categorised as a score of 23–50 or 0–22, based on the median split; as such, a total of 452 (52.3%; 95%CI 48.9 to 55.7) were categorised as having a score of 23 to 50 and 412

**Table 1. Univariable and multivariable of factors associated with knowledge and practice scores (N = 864).**

| | | Univariable analysis | | Multivariable analysis | Univariable analysis | | Multivariable analysis |
|---|---|---|---|---|---|---|---|
| | | | | Knowledge score[a] | | | Practices score[b] |
| | Frequency (%) (N = 864) | Score 23–50 (n = 452) | p-value | Score 23–50 *vs* 0–22 OR (95% CI) | Score 40–47 (n = 455) | p-value | Score 40–47 *vs* 17–39 OR (95% CI) |
| *Socio-demographics* | | | | | | | |
| **Age group (years)** | | | | | | | |
| 18–30 | 226 (26.2) | 110 (48.7) | | 0.65 (0.41–1.02) | 99 (43.8) | | Reference |
| 31–50 | 342 (39.6) | 212 (62.0) | p<0.001 | 1.27 (0.83–1.93) | 166 (48.5) | p<0.001 | 1.44 (0.93–2.22) |
| 51–89 | 296 (34.3) | 130 (43.9) | | Reference | 190 (64.2) | | 4.26 (2.53–7.16)*** |
| **Gender** | | | | | | | |
| Male | 278 (32.2) | 148 (53.2) | 0.716 | | 136 (48.9) | 0.145 | |
| Female | 586 (67.8) | 304 (51.9) | | | 319 (54.4) | | |
| **Ethnicity** | | | | | | | |
| Malay | 445 (51.5) | 291 (65.4) | | 2.06 (1.34–3.15)** | 319 (71.7) | | 29.56 (15.57–56.09)*** |
| Chinese | 228 (26.4) | 73 (32.0) | p<0.001 | 0.74 (0.47–1.17) | 119 (52.2) | p<0.001 | 19.98 (9.87–36.51)*** |
| Indian | 185 (21.4) | 87 (47.0) | | Reference | 15 (8.1) | | Reference |
| Others[†] | 6 (0.7) | 1 (16.7) | | - | 2 (33.3) | | - |
| **Highest education level** | | | | | | | |
| Primary school and below | 124 (14.4) | 41 (33.1) | | Reference | 54 (43.5) | | Reference |
| Secondary school | 312 (36.1) | 130 (41.7) | p<0.001 | 1.37 (0.83–2.26) | 152 (48.7) | 0.004 | 1.56 (0.88–2.78) |
| Tertiary | 428 (49.5) | 281 (65.7) | | 2.48 (1.41–4.34)** | 249 (58.2) | | 1.99 (1.02–3.91)* |
| **Average monthly household income (MYR)** | | | | | | | |
| <RM1000 | 221 (25.6) | 87 (39.4) | | Reference | 98 (44.3) | | Reference |
| RM1001-3000 | 413 (47.8) | 223 (54.0) | p<0.001 | 1.15 (0.78–1.73) | 227 (55.0) | 0.015 | 1.61 (1.04–2.50)* |
| >RM3000 | 230 (26.6) | 142 (61.7) | | 1.28 (0.78–2.10) | 130 (56.5) | | 1.38 (0.82–2.34) |
| **Occupation category** | | | | | | | |
| Professional and managerial | 182 (21.1) | 130 (71.4) | | 1.45 (0.86–2.43) | 96 (52.7) | | |
| General worker/ Self-employed | 239 (27.7) | 111 (46.4) | p<0.001 | 0.72 (0.46–1.13) | 110 (46.0) | | |
| Housewife | 223 (25.8) | 107 (48.0) | | 0.86 (0.54–1.37) | 127 (57.0) | 0.087 | |
| Student/ Retired/ Unemployed | 220 (25.5) | 104 (47.3) | | Reference | 122 (55.5) | | |
| **Region[††]** | | | | | | | |
| Northern | 142 (16.4) | 56 (39.4) | | Reference | 87 (61.3) | | 1.71 (0.64–4.56) |
| Southern | 259 (30.0) | 124 (47.9) | p<0.001 | 1.52 (0.96–2.42) | 113 (43.6) | | 0.68 (0.27–1.74) |
| Central | 155 (17.9) | 73 (47.1) | | 1.65 (0.99–2.76) | 61 (39.4) | p<0.001 | 0.72 (0.27–1.90) |
| East Coast | 275 (31.8) | 183 (66.5) | | 2.28 (1.69–4.54)*** | 177 (64.4) | | 0.77 (0.30–1.99) |
| Borneo Island | 33 (3.8) | 16 (48.5) | | 1.66 (0.73–3.75) | 17 (51.5) | | Reference |
| **Area of living** | | | | | | | |
| Urban | 405 (46.9) | 223 (55.1) | | 1.95 (1.24–3.08)** | 193 (47.7) | | Reference |
| Suburban | 279 (32.3) | 156 (55.9) | 0.002 | 1.74 (1.12–2.68)* | 157 (56.3) | 0.020 | 0.87 (0.58–1.31) |
| Rural | 180 (20.8) | 73 (40.6) | | Reference | 105 (58.3) | | 0.86 (0.52–1.40) |
| *Antibiotic and antibiotic resistance related knowledge* | | | | | | | |
| **Total knowledge score** | | | | | | | |
| 0–22 | 412 (47.7) | | | | 154 (37.4) | p<0.001 | Reference |
| 23–50 | 452 (52.3) | | | | 301 (66.6) | | 3.94 (2.71–5.73)*** |

[†]Excluded from multivariable analysis due to small sample size.

[††]Central: Selangor, Kuala Lumpur, Putrajaya, Negeri Sembilan; Southern: Johor, Malacca; Northern: Perlis, Kedah, Pulau Pinang, Perak; East Coast: Terengganu, Kelantan, Pahang; Borneo Island: Sabah, Sarawak, Labuan.

*p<0.05,

**p<0.01,

***p<0.001.

[a]Hosmer–Lemeshow test, chi-square: 7.490, p-value: 0.485; Nagelkerke $R^2$: 0.229.

[b] Hosmer–Lemeshow test, chi-square: 7.691, p-value: 0.464; Nagelkerke $R^2$: 0.448.

(47.7%; 95%CI 44.3 to 51.1) were categorised as having a score of 0–22. Higher score range indicates a higher level of knowledge. Table 1 shows the multivariable logistic regression analysis of demographics factors associated with having a higher knowledge score. Significantly higher knowledge scores were reported among the participants of the Malay ethnicity, whereas the lowest scores were found among the Chinese. There was a significant gradual increase in the level of knowledge as the level of educational attainment increased. By geographical region, participants from the East Coast and urban or suburban areas tended to have significantly higher knowledge scores. Higher average household income was found to be significantly associated with a higher level of knowledge in the univariate analyses, however the association was not significant in the multivariable model.

## Practices regarding antibiotic use

S2 Fig shows the findings of practices relating to antibiotic use. Regarding practices of obtaining antibiotics, a small proportion reported ever obtaining antibiotics from a pharmacy without a physician prescription (18.3%, n = 158) and seeking alternative doctors in obtaining antibiotics if a doctor refused to prescribe antibiotics (10.9%, n = 94). Malpractice in adherence to recommended doses (25.0%, n = 216) and using a lower (8.2%, n = 71) or higher (6.7%, n = 58) antibiotic dose than recommended were reported. A considerable proportion reused leftover antibiotics from previous treatments (19.4%, n = 168), discarded leftover antibiotics (17.2%, n = 149), shared leftover antibiotics with other people (14.4%, n = 124), or gave leftover antibiotics to pets (5.4%, n = 47).

The mean for the antibiotic practice scores was 37.9 (SD ± 6.5; range 17 to 47) out of a possible score of 48. The median was 40.0 (interquartile range [IQR], 35.0 to 42.0). Likewise, the significant value of the Kolmogorov-Smirnov test was less than 0.05 implies the data were not normally distributed. The scores were categorised as a score of 40–47 or 17–39, based on the median split; as such, a total of 455 participants (52.7%; 95%CI 49.3 to 56.0) were categorised as having a score of 40 to 47 and 409 participants (47.3%; 95%CI 44.0 to 50.7) were categorised as having a score of 17–39. Higher score range indicates a higher level of appropriate antibiotic practices.

In the multivariable model (Table 1), by demographics, the age group 51–89 years was more likely to have high antibiotic practice scores (OR 4.26, 95%CI 2.53–7.16) compared to the youngest age group (18–30 years). Participants of Malay (OR 29.56; 95% CI 15.57–56.09) and Chinese (OR 19.98; 95% CI 9.87–36.51) ethnic groups were more likely to have higher antibiotic practice scores than Indians. Participants who earned an average monthly household income of MYR1001-3000 (OR 1.61, 95% CI 1.04–2.50) were more likely to have a higher antibiotic practices score than those earning <MYR1000. Participants with tertiary education attainment showed higher practice scores (OR 1.99; 95%CI 1.02–3.91) than those with education levels of primary school and below. Higher antibiotic knowledge scores (OR 3.94; 95% CI 2.71–5.73) were associated with higher practice scores. The result of mediation analysis showed that age mediates the association between knowledge and antibiotic practices (β = 0.251; 95% CI: (0.131, 0.221).

## Discussion

This study describes the knowledge and practices regarding antibiotics and antibiotic resistance among the general Malaysian public. The study identified crucial gaps in knowledge and practices that provide important information for the development of an educational intervention to fill the identified gaps.

Our results show that the majority scored below the mid-point of the knowledge scale, implying a generally low level of knowledge about antibiotics and antibiotic resistance among the study participants. In particular, the study revealed the widespread misperception that antibiotics are effective for treating viral infections. Erroneous beliefs that antibiotics could speed up recovery of coughs and colds were also prevalent. The findings of this study are in concordance with previous small-scale findings in Malaysia that similarly reported that most of the participants viewed antibiotics as effective against viral infections [9, 16, 17]. Such misunderstandings can lead to the suboptimal use of antibiotics. Therefore, it is important to enlighten the public that viral infections do not respond to antibiotic treatment. As many viewed common cold and flu as infections that can be treated with antibiotics, it is equally important to educate the public on the differences between bacterial and viral infections. As evident in this study, knowledge regarding certain diseases that can or cannot be treated with antibiotics was prevalent. Educational interventions were found to have a significant effect in changing knowledge, attitudes and the public's appropriate use of antibiotics [18], which should be implemented in the Malaysian context.

Studies have shown that awareness about antibiotic resistance and the factors responsible for it remain largely unrecognised at the population level [19, 20]. Likewise, in this study, the terms drug resistance, antimicrobial resistance, and superbugs had not been heard of by the majority of participants. The lay public may not understand the complex mechanisms leading to antibiotic resistance and the consequences of their over or misuse. Evidence indicates that awareness campaigns directed to the general population have led to a substantial reduction in prescribing [20]. To date, there is a lack of public awareness-raising initiatives that focus on sensitising the public on the rational consumption of antibiotics in Malaysia. Henceforth, it is time to implement a campaign to enhance the public's understanding of the cause of antibiotic resistance and its serious consequences in order to promote the appropriate use of antibiotics.

As for findings of demographic factors influencing knowledge, this study found that having higher education attainment was associated with a higher level of knowledge, which was similarly found in many other studies worldwide [21–23]. The gradual increase in the level of knowledge as the educational level increase implies that educational intervention should be targeted at the less educated segment of the population [24]. Ethnic disparities in knowledge were evident. The low level of knowledge among the Chinese participants found in this study warrants further investigation. Higher levels of knowledge among participants from urban and suburban compared to those in rural regions indicates the need for focused and targeted interventions targeting remote and rural areas of Malaysia.

With regard to practices on antibiotic use, the median score of 40 out of a maximum score of 48 implies generally good practices regarding antibiotic usage. Despite these, there are some obvious practice gaps revealed in this study. It is worrisome that some participants obtained antibiotics from pharmacy without a physician's prescription. Antibiotics dispensed without a prescription are largely the cause of antibiotic misuse and overuse leading to antibiotic resistance [25]. In Malaysia, antibiotics are prescription-only medicines. More stringent inspections by the regulatory authorities are needed to avoid dispensing antibiotics without a prescription. Furthermore, some reported seeking an alternative doctor in obtaining antibiotics if a doctor refused to prescribe antibiotics, which indicates the need for sensitisation and training of physicians to practice rational antibiotic prescribing [26].

It is of utmost importance that the general public should practice rational utilisation of antibiotics as non-adherence to antibiotic consumption may increase the risk of drug resistance. In this study, malpractice such as non-adherence to antibiotic dosage schedules was reported by a considerable proportion of participants. Non-compliance may result in a large portion of leftover antibiotics. It has been reported in the literature that approximately 50% of antibiotics

used in self-medication were leftover [27]. In this study, nearly 20% reported ever using leftover antibiotics from previous treatments or sharing leftover antibiotics with others, implying the need to educate the public about understanding the importance of compliance to recommended medication plans and the danger of misuse of leftover antibiotics, including on humans and pets. The public should be enlightened that giving leftover antibiotics to pets can result in antibiotic-resistant infections in pets, which can potentially be passed on to a human.

Other hazardous practices such as throwing away leftover antibiotics and giving leftover antibiotics to pets reported in this study are also of high concern. Likewise, a study conducted in Malaysia reported that the majority of patients discard unused or expired medication in the garbage or flush them down the toilet; only a minority returned expired or unused medication to pharmacies [28]. Improper disposal of unused and expired antibiotics can lead to antibiotics leaching into the water system, causing microbes to mutate into resistant pathogens, promoting antimicrobial resistance, and resulting in environmental hazards and public health risks [29]. A policy aimed at preventing the disposal of antibiotics as household waste should be developed and implemented. Patients often lack knowledge on how to safely dispose of their leftover medications. Physicians or pharmacists should educate patients during prescription about the importance of medication adherence and proper antibiotic disposal during dispensing. Although some hospitals may have their own guidelines [30], there is a lack of national guidance on return unwanted medications. In 2010, the Pharmaceutical Services Division, Ministry of Health Malaysia (MOH) implemented the Return Your Medicines Program encouraging patients to return their unused or excess medicines to the pharmacy counter or medicine return box provided at the pharmacy facilities in the MOH hospitals and health clinics [31]. Patients should be informed and made aware of such guidelines so that they know where to return their expired, unwanted, or unused antibiotics.

By demographics, people with higher education and higher income were generally more knowledgeable about appropriate antibiotic use and antimicrobial resistance [32, 33]. Likewise, in this study, higher education and income were associated with better antibiotic usage. There is a gradual increase in the appropriate use of antibiotics as income level increases. This finding can be attributed to the fact that a population with a low socioeconomic status has greater barriers in accessing the health system, which discourages consultations and promotes self-medication [34, 35]. Therefore, education targeting poor households should be implemented to minimise misuse. Both the Chinese and Malay participants exhibited a higher level of appropriate antibiotic use than the Indian participants. These disparities imply that there is a need to understand if the knowledge, attitudes, and practices of antibiotic use are deeply rooted in a sociocultural milieu that varies from one ethnic community to another. However, it should be noted that the number of Chinese and Indian participants in this study is relatively smaller than the Malays. There is a need to further explore this in future research.

An important highlight of this study is the association between knowledge and practices related to antibiotic use. This study adds to the evidence that better knowledge increased the appropriate use of antibiotics. As found in other study, better knowledge was found to be associated with more appropriate use of antibiotics [36]. As antibiotic use is closely related to knowledge, our findings call for communication interventions that target the general public to fill the knowledge gaps identified in this study. The result of mediating effect of age on the association between knowledge and antibiotic practices along with the finding of higher odds of positive antibiotic practices score in older age participants imply that individuals of a younger age should be the target population for antibiotic use interventions. Future research is warranted to determine the effectiveness of antibiotic use intervention as well as to measure its cost-effectiveness to provide insights into effective strategies to promote prudent use of antimicrobials in the country.

This study has some strengths and limitations. First, is the limitation of the utility of telephone interviews, whereby all information obtained from the interview was self-reported and possibly be subjected to social desirability response and self-report bias. A second limitation is related to the CATI method, which only included households with fixed-line telephones; consequently, households without a telephone line were under-represented. Of note, increasing numbers of households no longer use landline phone service and instead rely on mobile phones to stay connected. The third limitation is the use of self-developed scales for the measurement of antibiotic and antibiotic resistance-related knowledge. Only the internal consistency of the scale was examined. Extensive psychometric analyses were not carried out. Therefore, the measurements using these scales should be interpreted cautiously. Lastly, despite this study is a nationwide survey that collected data from all states in Malaysia, some aspects of the cross-sectional design limit the generalization of our findings. The response rate was higher in females compared to males. Further, the income level of the majority of our study respondents was below the average monthly household income of the Malaysian population. Of note, Malaysia's median income was recorded at RM5,873 in the year 2019 [37]. Hence, our study may have a slight representation of respondents from the lower-income groups. Additionally, the ethnic group distribution in our study is slightly different from the general Malaysian population, of which 67.4% were Malays and Bumiputera, 24.6% Chinese, 7.3% Indians, and 0.7% Others [38]. Despite these limitations, the study was the first nationwide survey in Malaysia to evaluate antibiotic knowledge and practices that was carried out on a wide spectrum of socio-demographics.

## Conclusion

This study has documented important gaps in knowledge on antibiotic malpractices among the Malaysian public. This implies that enhancing health literacy associated with antibiotic use, knowledge, and awareness of antimicrobial resistance is a public health priority. Areas of focus that need to be addressed when developing an educational intervention to increase their knowledge and change practices have been identified. Socioeconomic disparities in knowledge and practices exist and addressing these inequalities should be the priority in future interventions. These findings will help health policymakers in Malaysia to implement target-specific education interventions to enhance antimicrobial resistance health literacy and practices.

## Supporting information

**S1 File. Survey questionnaire.**
(PDF)

**S1 Fig. Proportion of correct responses on knowledge items (N = 864).**
(TIF)

**S2 Fig. Proportion of ever practice (*rarely, occasionally, sometimes* and *often*) (N = 864).**
(TIF)

## Acknowledgments

We thank Department of Veterinary Services (DVS) and the Department of Fisheries (DOF), Malaysia and One Health University Network (MyOHUN) for assistance in questionnaire development.

## Author Contributions

**Conceptualization:** Li Ping Wong, Suraya Amir Husin, Zawaniah Brukan Ali, Benedict Sim, Sasheela Sri La Sri Ponnampalavanar.

**Data curation:** Haridah Alias, Suraya Amir Husin, Zawaniah Brukan Ali, Benedict Sim, Sasheela Sri La Sri Ponnampalavanar.

**Formal analysis:** Li Ping Wong, Haridah Alias.

**Funding acquisition:** Sasheela Sri La Sri Ponnampalavanar.

**Investigation:** Li Ping Wong.

**Methodology:** Li Ping Wong, Suraya Amir Husin, Zawaniah Brukan Ali, Benedict Sim, Sasheela Sri La Sri Ponnampalavanar.

**Project administration:** Li Ping Wong, Sasheela Sri La Sri Ponnampalavanar.

**Writing – original draft:** Li Ping Wong.

**Writing – review & editing:** Li Ping Wong.

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
