## [Decision Letter · Decision Letter 0]

16 Jul 2021

PONE-D-21-20468

Factors influencing inappropriate use of antibiotics: findings from a nationwide survey in Malaysia

PLOS ONE

Dear Dr. Wong,

Thank you for submitting your manuscript to PLOS ONE. After careful consideration, we feel that it has merit but does not fully meet PLOS ONE’s publication criteria as it currently stands. Therefore, we invite you to submit a revised version of the manuscript that addresses the points raised during the review process.

Kindly revise your manuscript according to the reviewers' comments and the revised manuscript will be reviewed again. 

A rebuttal letter that responds to each point raised by the academic editor and reviewer(s). You should upload this letter as a separate file labeled 'Response to Reviewers'.A marked-up copy of your manuscript that highlights changes made to the original version. You should upload this as a separate file labeled 'Revised Manuscript with Track Changes'.An unmarked version of your revised paper without tracked changes. You should upload this as a separate file labeled 'Manuscript'

We look forward to receiving your revised manuscript.

Kind regards,

Pathiyil Ravi Shankar

Academic Editor

PLOS ONE

Journal Requirements:

2. Please include additional information regarding the survey or questionnaire used in the study and ensure that you have provided sufficient details that others could replicate the analyses. For instance, if you developed a questionnaire as part of this study and it is not under a copyright more restrictive than CC-BY, please include a copy, in both the original language and English, as Supporting Information. Moreover, please include more details on how the questionnaire was pre-tested, and whether it was validated. 

3. Please amend your current ethics statement to address the following concerns:  

a) Did participants provide their written or verbal informed consent to participate in this study? 

b) As consent was verbal, please explain i)  how you documented participant consent, and ii) whether the ethics committees/IRB approved this consent procedure.

5. Please upload a new copy of Figures 1 and 2 as the detail is not clear. Please follow the link for more information: https://blogs.plos.org/plos/2019/06/looking-good-tips-for-creating-your-plos-figures-graphics/" https://blogs.plos.org/plos/2019/06/looking-good-tips-for-creating-your-plos-figures-graphics.

Additional Editor Comments (if provided):

Dear authors

Thank you for subnmitting your manuscript. Kindly carry out the revisions required by the reviewers and resubmit your manuscript for a second round of review.

Reviewers' comments:

Reviewer's Responses to Questions

**Comments to the Author**

1. Is the manuscript technically sound, and do the data support the conclusions?

Reviewer #1: Partly

Reviewer #2: Partly

2. Has the statistical analysis been performed appropriately and rigorously? 

Reviewer #1: No

Reviewer #2: No

3. Have the authors made all data underlying the findings in their manuscript fully available?

Reviewer #1: No

Reviewer #2: No

4. Is the manuscript presented in an intelligible fashion and written in standard English?

Reviewer #1: Yes

Reviewer #2: No

5. Review Comments to the Author

Reviewer #1: 1. The actual questionnaire is not available.

2. This is a newly developed questionnaire but the survey questions were only partially validated. I suggest the authors assess the validity and reliability of the questionnaire by test-retest assessment, inter-rater analysis, using principal component analysis, then check the internal consistency of questions loading onto the same factor.

3. Please state the supporting resources provided by the WHO.

4. "Sg Buloh Hopsital" was misspelled. It should be "Sg. Buloh Hospital".

5. The introductory sentence in the Abstract is misleading, readers may think that this study is assessing the threat of antimicrobial resistance in Malaysia.

6. In the "Introduction" section, reference 5 (2013) is out of date. Please provide updated information.

7. In the "Instruments" section, individuals may interpret “rarely”, “occasionally”, “sometimes”, “often” differently. The scales between “rarely”, “occasionally”, “sometimes”, “often” are not linear. How did the researchers address this inter-rater or “inter-respondent” difference?

8. There are 5 response options in the practices of antibiotic use questions; namely, never, rarely, occasionally, sometimes, and often. How are the 4 score options of 0, 1, 2, and 3 awarded to 5 response options?

9. The "Statistical analyses" section is not clear. The authors may describe clearly how the respondents were divided into 2 groups for comparisons.

10. In 2019, the average monthly household income in Malaysia is RM 7,901. About 75% of the respondents in the study fall under the bottom 20% of the classification of household income of the Malaysian community. Would the respondents be representative of a general Malaysian population?

11. The racial distributions in Malaysia are Malays and indigenous peoples 62%, Chinese 20.6%, and Indian 6.2%. How well is the racial distribution of Malaysia reflected in the racial distribution of the respondents in this survey?

12. The official retirement age in Malaysia is 60 years. Is there any particular reason that the age of the respondents was categorized as 18-30, 31-50, and 51-89?

13. What is the rationale for categorizing "students", "retired", "unemployed" as a group while they have a diverse age range, income, and experience?

14. Table 1, last section, the label for data "154 (37.4) p<0.001 Reference" is missing.. Table 1, last section, the label for data "154 (37.4) p<0.001 Reference" seems missing.

15. In the "Discussion" section, the authors state that better knowledge was found to be associated with more appropriate use of antibiotics. There seems to be a mismatch between the knowledge scores and the practice scores especially among the Indian race. The Indians who had a reasonable score (47% attaining a knowledge score range of 23-50) had very low practice scores (92% with a scores range of 17-39). Does it imply that an arbitrary cut-off point at the median score does not reflect the true underlying picture? Or Are the knowledge questions truly measure what they are supposed to measure?

16. Also, different individuals have different perceptions on the terms rarely, occasionally, sometimes, and often. The “distance” or “scales” between these items are very arbitrary and not constant. Would the authors discuss quantifying these terms with a continuous number?

17. The resolution of Figure 1 and Figure 2 is poor.

18. There are some minor typographical and grammatical errors in the manuscript.

Reviewer #2: Thank you for giving me the chance to review this ms. Here are my comments:

1. authors have noted that - This study has documented important gaps in knowledge on antibiotic malpractices among

the Malaysian public - in my opinion it is not new! even the refs 9-13 have indicated that others have done such studies.

2. The Centre for Epidemiology and Evidence-Based Practice, Department of Social and Preventive

Medicine should put its energy, money, time and effort to study the approach to improve and coming out with evidence-based interventions e.g. cost-effective interventions. The consequences and reasons for misuse are well documented, including in LMICs. So, what is the benefit of studying again and again ? We should find ways to overcoming these problems.

3. Methods - are poorly written and structured. study design? sample size? sampling from the 16 areas? Malaysian is a multiracial society with different culture, belief and socio-economic status - are these factors been taken into consideration during sampling? analysis? any confounders?

4. How effective is tel interview in Malaysia in getting a comprehensive views of the respondents?

5. Inclusion criteria - e.g. had heard - if just heard but have not used before, do you think there will be a different in their responses, understanding, knowledge and practice?

6. Tools and items - source of these items and questions? Your own ideas? or from the literature?

7. Translations - the process of translations? validations? lack of explanation, vague

8. Interviewers - was the interrater reliability measured? how did you reduce potential bias?

9. Cronbach's alpha - move to instrument development section

10. Analysis section - lack of information - univariate analysis? descriptive stats? how was the factors decided into the model, i.e. multivariate analysis?

11. all % must be supplied with n i.e. (n, %)

12. Your age category - 51-89 - in this category - how do you explain and separate the elderly population's knowledge and practice?

13. Discussion - what does the study add? what are the study findings implication?

14. Figures 1 and 2 are blur and cannot be read

15.This ms is better to be submitted in a local journal.

6. PLOS authors have the option to publish the peer review history of their article (what does this mean?). If published, this will include your full peer review and any attached files.

Reviewer #1: **Yes: **YokeLin LO

Reviewer #2: No

---

## [Author Response · Author response to Decision Letter 0]

29 Jul 2021

We have addressed the reviewers’ comments and upload the file labeled 'Response to Reviewers' and manuscript with track changes in the submission.

---

## [Decision Letter · Decision Letter 1]

1 Sep 2021

PONE-D-21-20468R1

Factors influencing inappropriate use of antibiotics: findings from a nationwide survey in Malaysia

PLOS ONE

Dear Dr. Wong,

Thank you for submitting your manuscript to PLOS ONE. After careful consideration, we feel that it has merit but does not fully meet PLOS ONE’s publication criteria as it currently stands. Therefore, we invite you to submit a revised version of the manuscript that addresses the points raised during the review process.

Dear authors

The reviewers have recommended some further clarifications about your manuscript. You are invited to carry out these changes. 

We look forward to receiving your revised manuscript.

Kind regards,

Pathiyil Ravi Shankar

Academic Editor

PLOS ONE

Journal Requirements:

Reviewers' comments:

Reviewer's Responses to Questions

**Comments to the Author**

1. If the authors have adequately addressed your comments raised in a previous round of review and you feel that this manuscript is now acceptable for publication, you may indicate that here to bypass the “Comments to the Author” section, enter your conflict of interest statement in the “Confidential to Editor” section, and submit your "Accept" recommendation.

Reviewer #2: All comments have been addressed

Reviewer #3: (No Response)

Reviewer #4: All comments have been addressed

2. Is the manuscript technically sound, and do the data support the conclusions?

Reviewer #2: Yes

Reviewer #3: Yes

Reviewer #4: Yes

3. Has the statistical analysis been performed appropriately and rigorously? 

Reviewer #2: Yes

Reviewer #3: Yes

Reviewer #4: Yes

4. Have the authors made all data underlying the findings in their manuscript fully available?

Reviewer #2: Yes

Reviewer #3: Yes

Reviewer #4: Yes

5. Is the manuscript presented in an intelligible fashion and written in standard English?

Reviewer #2: Yes

Reviewer #3: Yes

Reviewer #4: Yes

6. Review Comments to the Author

Reviewer #2: Thank you for taking seriously the reviewers' comments and suggestions. The revised ms has improved its quality.

Reviewer #3: Title: Factors influencing inappropriate use of antibiotics findings from a nationwide survey in Malaysia

I would like to thank the Editor for giving me the opportunity to review this manuscript.

Overall, it is a well written manuscript and addresses an important area of medicine use. My specific comments are mentioned below:

Title: I suggest adding ‘general public’ in the title

Abstract:

Line 21, Misuse of antibiotic………, is not correct. There are other studies (https://bmcinfectdis.biomedcentral.com/articles/10.1186/s12879-016-1530-2). Hence I suggest authors to modify this sentence.

Line 32, ‘low awareness’ what do authors mean by this? These should be a cutoff score mentioned to categorize as ‘low’, ‘average’, ‘high etc.’.

Introduction:

This section is generally well written. However, it lacks thorough literature review. I suggest authors to present more literature from Malaysia and from other countries.

A thorough justification for the research is lacking. Authors should describe similar studies from Malaysia and justify why this research was needed.

This section should add information for international readers. What it adds to international readers?

Line 61-63 shows similarity with https://www.hindawi.com/journals/ipid/2018/8492740/?

I suggest authors to run similarity check and modify it.

Materials and methods:

The study timeline should be mentioned. Was it conducted during COVID 19 pandemic? Was there any stratification for samples? How do we ensure the samples are equally distributed in terms of gender, education, income, ethnicity, geographic distribution etc. I could see the females are much more than males? Can this influence the study findings? Do the selected samples reflect Malaysian population nationwide? If not this should be mentioned as a study limitation.

Instruments

Line 109, the knowledge scores should be categorized as low, average, and high. What is the cutoff score to category as ‘high’? The same has to be done for practice scores also. It would be more logical to divide the scores as low, average and high.

Line 117-119 is not enough. A more detailed description on questionnaire development should be added. Did authors obtain some information from published literature? In that case those should be cited.

Line 92-96, shows similarity with https://link.springer.com/article/10.1186/s12889-016-3409-y

Authors copied verbatim and hence needs to be edited.

Results:

Line 181, How did authors arrive at ‘high scores’ ? How were the scores categorized?

Line 191, On what basis did authors perform mean and median scores? Did authors perform a normality check? If yes, mention the name of the test with p value in the methodology section.

Line 194-195, this information should be mentioned in the methodology and references should be provided for median split

Line 217, was the normal distribution performed for practice scores?

Discussion:

This section is well written. It can be further improvised by discussing more on policy recommendations based on the research findings.

Lines 241- 255. There is nothing new in this part. Antibiotics in viral infections is a well-known fact.

Line 254, …..public’s antibiotic stewardship behavior’ needs amendment. Try to rephrase with more suitable words.

Limitations:

Line 340, Strengths and limitations can be in two different paragraphs.

Line 356, I suggest authors to perform thorough literature review and see whether their claim is right.

Figure 1, I had problem in viewing the Figure 1. The font size is too small. I suggest authors to check this.

Reviewer #4: Authors have article on Knowledge about antibiotic use and misuse using telephonic interviewing Malaysia . Due to widespread antibiotic misuse antibiotic resistance is increasing & multi drug resistant pathogen are common .

Topic is very relevant and worth studying and results show lack knowledge among low socioeconomic strata & educational level as reason for decreased awareness.

Study was done during COVID Pandemic whether Pandemic has affected survey results? Any specific points of lockdown related rise in misuse=se & stress related response is not clear. This can be clarified.

Telephonic survey results could be inferior to personal questionnaire based study

7. PLOS authors have the option to publish the peer review history of their article (what does this mean?). If published, this will include your full peer review and any attached files.

Reviewer #2: **Yes: **Mohamed Izham Mohamed Ibrahim

Reviewer #3: No

Reviewer #4: **Yes: **Mukhyaprana Manuru Prabhu

---

## [Author Response · Author response to Decision Letter 1]

6 Sep 2021

We have further made all changes as suggested and enclosed the detail in tracked and also attached the specific reviewer responses in the submission.

---

## [Decision Letter · Decision Letter 2]

4 Oct 2021

Factors influencing inappropriate use of antibiotics: findings from a nationwide survey of the general public in Malaysia

PONE-D-21-20468R2

Dear Dr. Wong,

We’re pleased to inform you that your manuscript has been judged scientifically suitable for publication and will be formally accepted for publication once it meets all outstanding technical requirements.

Kind regards,

Pathiyil Ravi Shankar

Academic Editor

PLOS ONE

Additional Editor Comments (optional):

Reviewers' comments:

Reviewer's Responses to Questions

**Comments to the Author**

1. If the authors have adequately addressed your comments raised in a previous round of review and you feel that this manuscript is now acceptable for publication, you may indicate that here to bypass the “Comments to the Author” section, enter your conflict of interest statement in the “Confidential to Editor” section, and submit your "Accept" recommendation.

Reviewer #3: All comments have been addressed

Reviewer #4: All comments have been addressed

2. Is the manuscript technically sound, and do the data support the conclusions?

Reviewer #3: Yes

Reviewer #4: Yes

3. Has the statistical analysis been performed appropriately and rigorously? 

Reviewer #3: Yes

Reviewer #4: Yes

4. Have the authors made all data underlying the findings in their manuscript fully available?

Reviewer #3: Yes

Reviewer #4: (No Response)

5. Is the manuscript presented in an intelligible fashion and written in standard English?

Reviewer #3: No

Reviewer #4: Yes

6. Review Comments to the Author

Reviewer #3: My previous comments have been addressed adequately by the authors. The revised manuscript reads well.

Reviewer #4: Article can be accepted

Revi8ewer comments have be addressed

Limitations of telephonic survey commented could have been addressed in more systematic way

7. PLOS authors have the option to publish the peer review history of their article (what does this mean?). If published, this will include your full peer review and any attached files.

Reviewer #3: No

Reviewer #4: No

---

## [Editor Report · Acceptance letter]

11 Oct 2021

PONE-D-21-20468R2 

Factors influencing inappropriate use of antibiotics: findings from a nationwide survey of the general public in Malaysia 

Dear Dr. Wong:

I'm pleased to inform you that your manuscript has been deemed suitable for publication in PLOS ONE. Congratulations! Your manuscript is now with our production department. 

Kind regards, 

on behalf of

Dr. Pathiyil Ravi Shankar 

Academic Editor

PLOS ONE